# Interaction between Cigarette Smoke and Human Papillomavirus 16 E6/E7 Oncoproteins to Induce SOD2 Expression and DNA Damage in Head and Neck Cancer

**DOI:** 10.3390/ijms24086907

**Published:** 2023-04-07

**Authors:** Diego Carrillo-Beltrán, Julio C. Osorio, Rancés Blanco, Carolina Oliva, Enrique Boccardo, Francisco Aguayo

**Affiliations:** 1Instituto de Bioquímica y Microbiología, Facultad de Ciencias, Universidad Austral de Chile, Valdivia 5090000, Chile; diego.carrillo@uach.cl; 2Laboratorio de Oncovirología, Programa de Virología, Instituto de Ciencias Biomédicas (ICBM), Facultad de Medicina, Universidad de Chile, Santiago 8380000, Chile; 3Department of Microbiology, Institute of Biomedical Sciences, University of Sao Paulo, Sao Paulo 05508-900, Brazil; eboccardo@usp.br; 4Departamento de Biomedicina, Facultad de Medicina, Universidad de Tarapacá, Arica 1000000, Chile

**Keywords:** cancer, head and neck, human papillomavirus, cigarette smoke, SOD2, DNA damage

## Abstract

Even though epidemiological studies suggest that tobacco smoking and high-risk human papillomavirus (HR-HPV) infection are mutually exclusive risk factors for developing head and neck cancer (HNC), a portion of subjects who develop this heterogeneous group of cancers are both HPV-positive and smokers. Both carcinogenic factors are associated with increased oxidative stress (OS) and DNA damage. It has been suggested that superoxide dismutase 2 (SOD2) can be independently regulated by cigarette smoke and HPV, increasing adaptation to OS and tumor progression. In this study, we analyzed SOD2 levels and DNA damage in oral cells ectopically expressing HPV16 E6/E7 oncoproteins and exposed to cigarette smoke condensate (CSC). Additionally, we analyzed SOD2 transcripts in The Cancer Genome Atlas (TCGA) Head and Neck Cancer Database. We found that oral cells expressing HPV16 E6/E7 oncoproteins exposed to CSC synergistically increased SOD2 levels and DNA damage. Additionally, the SOD2 regulation by E6, occurs in an Akt1 and ATM-independent manner. This study suggests that HPV and cigarette smoke interaction in HNC promotes SOD2 alterations, leading to increased DNA damage and, in turn, contributing to development of a different clinical entity.

## 1. Introduction

Head and neck cancers (HNCs) represent a global health problem which affected ~830,000 subjects in 2020 worldwide, causing the death of more than 50% of them [1]. This cancer originates from different anatomical locations, such as the oropharynx, larynx, hypopharynx, mouth, lips, palate, and tonsils. Late diagnosis is a common problem in this type of cancer, accompanied by regional and distal metastases that worsen the patient’s prognosis [2]. The risk factors associated with HNC are smoking, frequent alcohol consumption, and high-risk human papillomavirus (HR-HPV) infection [3]. HR-HPV E6 and E7 oncoproteins can induce the degradation of the tumor suppressors p53 and pRb, respectively, contributing to cell immortalization and transformation [4,5]. HPV16 is the most frequent HR-HPV genotype detected in HNCs, with 90% prevalence in this tumor [6,7,8]. However, HR-HPV infection is not a sufficient condition for carcinogenesis, with additional co-factors being required. Even though epidemiological studies suggest that tobacco smoking and HR-HPV are mutually exclusive risk factors for HNCs, some subjects who develop this cancer are both HR-HPV positive and smokers, suggesting the possibility of interactions [9].

Cigarette smoke is considered one of the leading agents related to HNC, this single factor being the cause of 42% of deaths associated with this disease [10]. More than 70 compounds in cigarette smoke are considered carcinogenic and classified into two types, those that directly promote DNA damage and those that require to be metabolized by host enzymes to promote the damage [11]. Most epithelial cells in head and neck tissues can be directly exposed to cigarette smoke, favoring the possibility of functional interactions with HR-HPV [12]. Both cigarette smoke and HR-HPV are inducers of oxidative stress (OS) since they favor the production of reactive oxygen species (ROS) such as hydrogen peroxide, nitric oxide, and superoxide that alter the cellular redox balance [13,14,15]. These changes lead to modifications in gene expression to modulate redox-sensitive proteins, many of which have been involved in cancer [14,16,17]. The family of superoxide dismutase (SOD) catalyzes the dismutation of superoxide (O_2_^−^) into hydrogen peroxide (H_2_O_2_) and oxygen and is suggested to be a prognostic factors in HNC [18,19]. In particular, the manganese superoxide dismutase 2 (SOD2) protein has the highest antioxidant capacity in the mitochondria when compared to other SOD family members. Moreover, SOD2 is suggested to be a prognostic biomarker of oral cancer malignancy because SOD2 overexpression is associated with lymph node metastases [18]. However, the role of SOD2 in cancer is complex, as it presents a dichotomy. First, SOD2 downregulation can lead to O_2_^−^ accumulation, which is associated with increased DNA damage and cell proliferation during tumorigenesis [20]. Conversely, SOD2 positive regulation goes hand in hand with an improvement in the elimination of O_2_^−^ and intracellular H_2_O_2_ increase, which correlates with adaptation to OS and activation of signaling pathways involved in tumor progression [21,22,23,24]. Specifically, increased SOD2 levels have been described in smoker HNC subjects and hypopharyngeal cells expressing the HPV16 E6 oncoprotein [25,26].

In this study, we report for the first time that HPV16 E6 and E7 oncoproteins and cigarette smoke cooperate to regulate SOD2 levels and DNA damage in head and neck cancer cells. Furthermore, we report a positive correlation between SOD2 upregulation and HPV16 E6 expression at the transcriptional level in HNCs from Chilean patients.

## 2. Results

### 2.1. HPV16 E6 and E7 Promote SOD2 Expression in Oral Cells

Oral cells (SCC143) were transduced with the retroviral vectors pLXSNE6/E7 or pLXSN. E6 and E7 transcripts were detected only in those cells transduced with the pLXSNE6/E7 vector (Figure 1A). Furthermore, we compared the levels of E6 and E7 transcripts in SCC143 and UM-SCC-100 cells with those obtained from SiHa cells (cervical cancer cell line containing two copies of HPV16 per cell) (Figure 1B,C). We found that the levels of E6/E7 transcripts in transduced SCC143 cells were similar to those obtained in SiHa cells and therefore selected these cells for further assays. Subsequently, we analyzed the proliferative capacity of SCC143V and SCC143E6/E7 cells, using a BrdU assay to analyze DNA synthesis. This assay detected a significant increase in proliferative activity in SCC143E6/E7 cells, and this, in turn, is dependent on ATM activation (Figure 1D). Then, using an RTqPCR assay, we saw a significant increase in SOD2 transcripts in SCC143 E6/E7 cells (Figure 1E). Finally, using WB, we detected the levels of the P53, pRb, pATM, ATM, and SOD2 proteins. The results demonstrate the functional activity of E6 and E7 through the downregulation of P53 and pRb, respectively (Figure 1F). In addition, the upregulation of pATM and SOD2 by the HPV16 E6 and E7 oncoproteins was confirmed (Figure 1F). These data indicate that the HPV16 E6 and E7 oncoproteins are increasing SOD2 transcripts and protein levels from oral cells.

### 2.2. SOD2 Is Regulated by HPV16 E6 in an AKT1/ATM Independent Manner in Oral Cells

An assay with siRNA was performed to determine the relevant oncoprotein in the regulation of SOD2 levels. SCC143 E6/E7 cells were transfected with siRNA for E6 and E7 and scrambled control (SCR) for this assay. After 48 h post-transfection, SOD2 levels were analyzed by WB. In addition, WBs for p53 and pRb were included as indirect functionality control of the siRNAs. The results show p53 and pRb levels recovery when the E6 and E7 interferents are used, respectively. In addition, we detected that when using the E6 interferent of HPV16, the SOD2 levels decreased significantly (Figure 2A). On the other hand, we analyzed whether the increase in SOD2 levels was dependent on ATM activation, which is related to DNA damage repair and is activated by E6 and E7 oncoproteins. Therefore, SCC143 E6/E7 cells were treated with two concentrations of the specific inhibitor of ATM activation (KU55933) for 24 h. We observed no statistically significant differences in SOD2 levels when the inhibitor KU55933 was used (Figure 2B). In addition, we analyzed the role of PI3K/AKT signaling in SOD2 levels induced by HPV16 oncoproteins, as the extent of this pathway in SOD2 transcriptional regulation was previously reported [20]. Accordingly, SCC143 E6/E7 cells were treated with the PI3K-specific inhibitor (LY294002) for 24 h. Then we analyzed the levels of AKT, pAKT1S473, and SOD2 proteins by WB. First, we detected a decrease in the phosphorylated form of AKT1S473 when comparing SCC143 pLXSN with SCC143 pLXSN E6/E7, and there was no significant change in SOD2 levels when using the inhibitor LY294002 (Figure 2C). The data suggest that HPV16 E6 is relevant for inducing SOD2 levels in an ATM- and PI3K/AKT-independent manner.

### 2.3. HPV16 E6 and E7 Oncoproteins Together with CSC Induce an Increase in SOD2 Levels and DNA Damage in Oral Cells

Both cigarette smoke and HPV are risk factors for HNC and inducers of oxidative stress. Therefore, we hypothesize that HPV oncoproteins E6 and E7 and cigarette smoke may collaborate to induce SOD2 and genetic damage in oral cells. To assess this possible collaboration, we first evaluated non-toxic doses of CSCs in SCC 143 cells by using the MTS assay with different concentrations of the compound (Figure 3A). The results indicate the maximum non-lethal amount is 50 μg/mL of CSC. Then, to verify the functionality of the compound, we evaluated the cytochrome p450 (CYP1B1) transcript by RT-PCR and analyzed the activation of ERK and AKT1 by WB (Figure 3B,C). We observed a significant increase in the CYP1B1 transcript in SCC143 cells exposed to 10 μg/mL CSC. Moreover, CSC can upregulate the phosphorylated forms of ERK and AKT1 in SCC143 V and SCC143 E6/E7 cells. These data confirm the biological activity of CSC in this oral model. To determine the effect of HPV16 E6/E7 and CSC on SOD2 levels, SCC143 V and SCC143 E6/E7 cells were exposed to 10 and 50 ug/mL CSC for 24 h, using DMSO as a control (Figure 3D). The levels of pATM, ATM, Rb, and SOD2 proteins were then analyzed by WB. We observed a significant increase in SOD2 levels with 10 and 50 ug/mL CSC in both SCC143 V and SCC143 E6/E7 cells. However, a more substantial rise in SOD2 levels is observed in SCC143 E6/E7 cells with CSC at both concentrations. In addition, we detected a CSC-mediated increase in pATM and a CSC-mediated decrease in Rb protein (Figure 3A). The data suggest that both factors independently induce SOD2 levels, but CSC provides a more significant increase in SOD2 levels when HPV16 E6 and E7 are present. An immunofluorescence assay for Gamma-H2AX, a marker for single- and double-stranded DNA breaks, was performed to determine changes in genetic damage. In this way, SCC143 V and SCC143 E6/E7 cells were treated with 10 and 50 ug/mL of CSC for 24 h. Inhibitor KU55933 was included in this assay as a control to potentiate the damage induced by both factors. We observed that SCC143E6/E7 cells present a significant increase in the fluorescence intensity of Gamma-H2AX concerning the SCC143V. Furthermore, it can be seen that CSC induces a substantial increase in the fluorescence intensity of Gamma-H2AX compared to DMSO.

Furthermore, a more significant rise in Gamma-H2AX fluorescence intensity can be observed in CSC-treated SCC143E6/E7 cells. Upon exposing cells to ATM inhibitor and CSC, the fluorescence intensity increases due to the blockage of ATM-mediated repair (Figure 4). In summary, CSC with E6/E7 oncoproteins can individually induce SOD2 levels and genetic damage in oral cells. However, when both factors are present, the observed changes in SOD2 levels and genetic damage are enhanced.

### 2.4. HPV16-Positive HNSCCs Correlate with Upregulation of SOD2 Transcripts

We evaluated the levels of SOD2 transcripts in 49 FFPE samples by RT-PCR. The cases were then distributed into three groups: HPV16 positive, HPV positive for another genotype, and HPV negative. When comparing the data of each group, we found statistically significant differences between the HPV-negative and the HPV16-positive samples (Figure 5A). Subsequently, we evaluated the E6 and E7 transcripts levels of the HPV16-positive samples. Considering these results, we distributed the data in groups of high or low levels of E6 or E7 transcripts. We detected that SOD2 levels correlate with high levels of E6 transcripts and not with E7 levels, presenting statistically significant differences between those samples that show differences in E6 transcript levels (Figure 5B). These data suggest that HPV16 E6 may be involved in SOD2 upregulation in HNC.

On the other hand, we performed analyses with the UCSC Xena web source to analyze a functional and phenotypic correlational genomic dataset of head and neck tumors from the GDC TCGA database. In this context, we detected that SOD2 expression was increased only in oral floor cancer samples with a smoking history compared with HPV-negative oral floor cancer samples without a smoking history (Figure 6). Regarding the variables analyzed, we did not find the HPV genotype of the tumors positive for the virus.

## 3. Discussion

Cigarette smoke and HR-HPV are both potent inducers of carcinogenesis and tumor progression, modifying multiple pathways associated with the hallmarks of cancer [27,28,29]. Thus, both factors induce molecular signals through changes in gene expression of tumor suppressors and oncogenes in the cell [28,29]. In HNC, these molecular changes define the patient’s clinical characteristics; thus, those HPV-positive tumors are defined as a different entity compared to cancers associated with cigarette smoke and alcohol [30]. Furthermore, inside HNC, the floor of the mouth is a common site for developing cancer with a high prevalence of HPV16 [31]. On the other hand, OSCC (posterolateral tongue and floor of mouth) is strongly associated with smoking history [32]. It is essential to consider the initiation of the tumor process since HPV-positive patients require other factors to generate cancer [33].

The reported prevalence of HPV in HNC depends on the anatomical origin; thus, the prevalence for the oropharynx is 25–85%; for the larynx, 20–25%; and for oral cancer, 24–32% [34,35,36]. We previously detected an HPV prevalence of 61.2% in oropharyngeal tumors and 11% in oral squamous cell carcinoma (OSCC) from Chile [7,37]. The HR-HPV genotype with the highest prevalence detected is HPV16, with 80% positivity in positive HPV samples, which is consistent with that reported by other authors [36,38]. Interestingly, via a transcript analysis, we detected a statistically significant increase in SOD2 only in HPV16 positive cases when compared to HPV-negative cases. SOD2 expression studies in clinical models of tongue cancer have shown a significantly higher expression than in normal tissue samples. Furthermore, SOD2 expression was higher in late-stage (stages III and IV) than in early stage disease (stages I and II) [39]. During lung cancer chemotherapy, SOD2 overexpression increases resistance to the tyrosine kinase inhibitor anlotinib, used as a third line of treatment for patients with advanced NSCLC. Specifically, SOD2 promotes the inhibition of mitochondrial ROS production and the suppression of apoptosis [40]. In addition, SOD2 has a protective role against radiotherapy; it increases the malignant properties of tumor cells, such as invasion, migration, and anchorage-independent growth [19,41,42].

Termini L et al. demonstrated that the SOD2 protein was associated with the malignancy of cervical cancer (100% prevalence of HPV), considering it a possible biomarker of progression in this type of cancer [43]. However, by immunohistochemical analysis, Rabello S et al. suggested that the upregulation of SOD2 was independent of the presence of HPV16 and HPV18 in this type of cancer [44]. In this regard, when evaluating the E6 and E7 transcripts in the HNC samples, we observed a positive correlation between the levels of the HPV16 E6 transcript with the levels of SOD2, which in part reflects a possible relationship between the levels of SOD2 dependent on the expression of HPV16 E6. Along with the above, we detected the upregulation of SOD2 transcripts and protein in HPV16 E6, and E7 transduced oral cells. In addition, using siRNAs, we observed a dependence of SOD2 on the oncoprotein E6. In that sense, Cruz-Gregorio C et al. showed, in a hypopharyngeal cell model, that HPV16 E6 can promote mitochondrial metabolism, increasing mitochondrial protein levels, OS, and genetic damage [26].

We attempted to determine how HPV16-mediated SOD2 regulation occurs by evaluating the involvement of ATM and PI3K/AKT1. First, we analyzed whether ATM activation mediated by E6 and E7 oncoproteins could induce SOD2 levels since ATM is reportedly required for NF-κB-mediated SOD2 expression in the mammary epithelium [45]. However, we did not detect SOD2-level alterations in cells expressing HPV16 E6 and E7 oncoproteins when they were exposed to the KU55933 inhibitor. On the other hand, we analyzed the SOD2 dependence of the PI3K/AKT1 pathway in the SCC143E6/E7 model since it has been reported that this pathway can regulate transcription factors that bind to the SOD2 promoter, favoring its transcription, such as NF-κB and CREB [20,46,47]. However, we observed a decrease in AKT1^S473^ levels when comparing CSCC143V and SCC143E6/E7 cells. Thus, as expected, we did not observe changes in SOD2 levels when cells were treated with LY294002. Interestingly, the decrease in pAKT1 has been associated with increased mitochondrial metabolism, favoring the expression of related proteins such as SOD2 [48]. In this regard, we hypothesize that the E6-mediated regulation of SOD2 is dependent on p53 downregulation. Previous studies show that p53 degradation allows the release of the transcription factor SP1, which maintains a binding site near the SOD2 promoter that is important in gene transcription [20,49].

Cigarette smoke cooperates with HR-HPV to promote cervical cancer progression [12,50,51]. However, studies that address the interaction of both components in HNC are scarce because epidemiological studies suggest that HR-HPV and tobacco are mutually exclusive factors for HNC development [12]. We previously reported that the interaction between cigarette smoke and HR-HPV induces DNA damage and increases E6 and E7 oncoproteins’ expression in lung and cervical cancer models [52,53]. Other groups have determined that the interaction between cigarette smoke and HR-HPV can induce the expression of viral oncoproteins, favor viral infection and host genome integration, increase genetic damage, and modify the expression of relevant proteins in tumor progression, among other things [54,55,56,57,58]. Specifically, in OPSCC, positivity for HPV is a factor for good prognosis and better response to treatment [59]. However, through a retrospective study in patients with OPSCC who had exposure to cigarette smoke and were HPV positive, it was observed that the prognosis of the disease worsened, suggesting that this type of tumor is a new entity [60]. In the case of oral epithelium from US patients, it has been determined that those who have been exposed to tobacco are significantly associated with HPV16 infection [61]. Our results suggest that CSC induces the regulation of ERK1/2 and PI3K/AKT signaling pathways in oral cells independently of HPV16 E6 and E7 expression. In this regard, Si-Youngk et al. found that PI3K signaling levels do not change between smoking and non-smoking HPV-positive patients [62]. On the other hand, we detected that SOD2 levels and genetic damage are synergistically increased by HPV16 E6/E7 and CSC in oral cells. In this regard, SOD2 has been reported to be independently induced by cigarette smoke and HPV E6 and E7 oncoproteins [26,63]. The cigarette smoke/HPV cooperation may increase H_2_O_2_-mediated OS, increasing genetic damage and metastasis initiation [64,65]. We understand that, in this study, we did not find the underlying molecular mechanism of SOD2 regulation by HPV/cigarette smoke. We also know that it needs to be replicated in more head and neck cell models and in an in vivo model to demonstrate the mechanisms involved. Regarding the clinical samples analyzed in this study, it should be noted that they are from the oropharynx. In future research, other anatomical locations of the head and neck would have to be added. Future studies may molecularly address how this regulation occurs and the phenotypic changes that may be promoted by the synergistically induced regulation of SOD2 by CSC and HPV E6/E7.

Cigarette smoke and HPV16 are factors involved in the development and progression of cancer. Therefore, these factors may interact in head and neck epithelial cells and promote changes associated with cellular malignancy. Understanding how HPV/cigarette smoke cooperation can alter head and neck cells can allow us to predict the molecular changes associated with HNC induced by both factors. The data from this study suggest that the upregulation of SOD2 levels is mainly mediated by the HPV16 E6 oncoprotein in HNC. We show, in a model, the data found in this study (Figure 7). Furthermore, this is the first report of the cooperation between cigarette smoke and the HPV16 E6 and E7 oncoproteins inducing sod2 levels and genetic damage in oral cells. Finally, we hypothesize that the genetic damage caused by the collaboration of both factors may favor neoplastic development or favor tumor progression through the positive regulation of SOD2. Further determination of the combined effects of risk factors may provide new insights to improve the management of patients with HNC.

## 4. Materials and Methods

### 4.1. Cell Lines, Culture and Transductions

The SCC143 cell line (floor of mouth squamous cell carcinoma) and the UM-SCC-100 (Head and neck squamous cell carcinoma) cell line were obtained from the University of Pittsburgh [66]. GP + envAM-12 (CRL-9641™) retrovirus packaging cells were kindly donated by Dr. Enrique Boccardo, Institute of Biomedical Sciences, University of Sao Paulo, Sao Paulo, Brazil. Cells were incubated in Dulbecco’s Modified Eagle Medium (DMEM) (Gibco, Carlsbad, CA, USA) supplemented with 10% fetal bovine serum (FBS) (Hyclone, Fremont, CA, USA), with antibiotics (100 units/mL penicillin and 100 g/mL streptomycin), and maintained at 37 °C in a 5% CO_2_ atmosphere. For the subculture, cells were incubated with trypsin for 3–5 min and maintained with a new medium containing FBS (Hyclone, Fremont, CA, USA). SiHa (HTB-35™) cervical carcinoma cells were obtained from the American Type Culture Collection (ATCC; Manassas, VA, USA) and cultured in RPMI-1640 basal medium (Gibco, Carlsbad, CA, USA) supplemented with 10% heat-inactivated fetal bovine serum (FBS) (Hyclone, Fremont, CA, USA), 100 U/mL penicillin, 100 g/mL streptomycin, and 0.25 μg/mL amphotericin B (Gibco, Carlsbad, CA, USA). Cells were tested for mycoplasma contamination. Plasmids pLXSN and pLXSNHPV16E6/E7 were kindly donated by Dr. Massimo Tommasino, from the International Agency for Research on Cancer (IARC), Lyon, France. Retroviral transduction was performed with GP + envAM-12 packaging cells previously transfected with pLXSN or pLXSNHPV16E6/E7 plasmids for 24 h at 37 °C in an atmosphere containing 5% CO_2_ with lipofectamine 2000 (Invitrogen, Carlsbad, CA, USA), according to the manufacturer’s instructions. SCC143 cells and UM-SCC-100 were stably transduced and were then selected by 0.3 mg/mL Geneticin (GIBCO, Carlsbad, CA, USA).

### 4.2. Viability Assays (MTS)

SCC143 cells 5 × 10^3^ were grown in 96-well plates. After 24 h, cells were treated with cigarette smoke condensate (CSC) prepared from 1R4F reference (Murty Pharmaceutical, Lexington, KY, USA). A stock solution of 40 mg/mL was prepared, and the maximum working solution that did not affect the viability of the cells was 50 µg/mL and was incubated for 72 h. Viability was measured using the CellTitter 96^®^ aqueous non-radioactive cell proliferation assay kit (Promega, Madison, WI, USA), from which 20 µL of the reagent was added to each well, and the cells were incubated for 3 h. Finally, the absorbance was measured at 490 nm.

### 4.3. Real-Time Polymerase Chain Reaction (qPCR)

The qPCR was realized in a AriaMx Real-Time apparatus (Agilent, Santa Clara, CA, USA) in a 25 μL final volume. The components for qPCR were as follows: 12.5 μL 2X SYBR Green Mastermix (Bioline, London, UK), 7.5 μL nuclease-free water, and 1 μL cDNA template. The thermocycling conditions were as follows: 94 °C for 30 s, 58 °C for 20 s, and 72 °C for 20 s, for a total of 40 cycles. The relative copy number of each sample was calculated through the 2^−ΔΔCt^ method. All reactions were performed in triplicate.

### 4.4. Western Blot

Protein lysates obtained from SCC143 empty vector and E6/E7-transduced cells were extracted with RIPA 1X lysis buffer (Abcam, Cambridge, UK) containing protease and phosphatase inhibitor cocktail (Roche, Basel, Switzerland). Suspensions were centrifuged at 14,000× *g* for 15 min at 4 °C. The protein concentration was quantified with the PierceTM BCA protein assay kit (Thermo Scientific, Rockford, IL, USA). Next, 25 μg of total protein was loaded per well and separated by SDS-PAGE on 12% gels. Proteins were then transferred by electroblotting to Hybond-P ECL membranes (Amersham, Piscataway, NJ, USA), using a pH 8.3 Tris-glycine transfer buffer (20 mM Tris, 150 mM glycine, and 20% methanol) and a Trans–Blot^®^ SD semi–dry electrophoretic transfer cell (Bio-Rad, Hercules, CA, USA). Membranes were blocked in 5% bovine serum albumin/0.5% Tween-20 in Tris-buffered saline pH 7.6 (TBS) for 1 h at room temperature (RT) and then incubated overnight at 4 °C with primary antibodies against p53 (BD554294) (BD PharmigenTM, San Diego, CA, USA), pRb (ab24), pATM (ab36810), ATM (ab78), (Abcam, Cambridge, UK), pAKT1S473 (4060S) (Cell Signaling, Danver, MA, USA), β-actin (SC47778), SOD2 (SC 137254), AKT1(SC5298). ERK (SC514302), pERK (SC136521) (Santa Cruz Biotechnology, Inc., Dallas, TX, USA), and 1:1000 in TBS/Tween 20 (TBS–T20). After three washes in TBS–T20, the membranes were incubated either with Anti-Mouse IgG (BD Pharmingen; BD Biosciences, Heidelberg, Germany) or Anti-Rabbit IgG (Santa Cruz Biotechnology, Inc., Dallas, TX, USA) conjugated to HRP, diluted 1:1000 in BSA 5% blocking buffer for 1 h at RT. Membranes were washed three times for 15 min and revealed with the ClarityTM Western ECL detection reagent (Bio-Rad, Hercules, CA, USA) on a (ChemiDocTM Bio-Rad), according to manufacturer’s instructions.

### 4.5. Immunofluorescence

Transduced cells were grown to confluence in treated coverslips in 24-well plate, washed twice with 1× PBS (pH 7.4), dried, and then incubated for 5 min with cold acetone/methanol. Next, cells were incubated with 3% bovine serum albumin (BSA) for 1 h at room temperature, followed by incubation with a primary monoclonal anti-specific protein antibody diluted in 1× PBS (1:50), according to the manufacturer’s instructions. The fixed cells were washed three times for 5 min at room temperature and incubated with a secondary fluorescein isothiocyanate (FITC) (Santa Cruz biotechnology, Dallas, TX, USA). After three washes with 1× PBS, cells were incubated for 15 min with DAPI (Thermo Fisher Scientific, Waltham, MA, USA) and finally visualized in a fluorescence microscope.

### 4.6. Tissue Samples

We used previously collected 49 formalin-fixed, paraffin-embedded (FFPE) Oropharyngeal squamous-cell carcinomas (OPSCC) obtained between 2009 and 2020 from the José Joaquín Aguirre Clinical Hospital [37], University of Chile (Santiago, Chile). Each case was analyzed by a histopathologist. This study was approved by the Board of Directors of the Ethics Committee of the Hospital Clínico José Joaquín Aguirre, Universidad de Chile (Number 47-2019). The samples were characterized previously by Oliva C. et al., reporting a 61.2% (30/49) positivity for HPV [37]. HPV16 is the most prevalent genotype with 80% (24/30), followed by HPV6 with 10% (3/30), HPV33 with 6.7% (2/30), and HPV18 with 3.3% (1/30) [37].

### 4.7. FFPE RNA Extraction, cDNA Conversion, and RT-PCR

RNA purification was carried out using the High Pure RNA paraffin kit (Roche), following the manufacturer’s instructions. The RNA obtained was suspended in 30 μL of RNA paraffin kit elution buffer and stored at −80 °C until use. The cDNA was prepared with 100 ng of purified RNA, RNAsin 1 U/μL (Promega, USA), 1× buffer TR (Promega, USA), 10 μg/μL random primers (Promega, USA), 20 U/μL MMLV (Promega, USA), and 2 mM dNTPs in a final volume of 20 μL. MMLV negative controls were included. The reaction mixture was incubated at 37 °C for 1 h and stored at −20 °C. RT-PCR was carried out using the primers of Table 1. β-actin mRNA levels were used for normalization of RNA expression. The amplification conditions were 94 °C for 5 min, followed by 33 cycles of denaturation at 95 °C for 45 s, annealing at 56 °C for 40 s, and extension at 72 °C for 45 s, with a final extension for 5 min at 72 °C. For semi-quantitative analysis, ImageJ software version 1.52a (National Institutes of Health, Bethesda, MD, USA) was used.

### 4.8. Gene Expression Analysis and Statistical Analysis

The UCSC Xena web source did allow us to explore functional genomic data sets and correlational genomic and phenotypic variables. We selected 613 HNSSC samples from GDC TCGA Head and Neck Cancer Database. We selected only 28 HNSSC that met the following criteria: (1) HPV status information, (2) tobacco-smoking history, (3) SOD2 expression, and (4) tumor primary site (floor of mouth).

A Mann–Whitney test was used to compare the means between two groups. Comparisons between multiple groups were performed using one-way ANOVA and Tukey’s post hoc test. All statistical tests were performed as two-sided and considered significant at a *p*-value < 0.05. Statistical analyses were run using the GraphPad Prism 6 software.

## Figures and Tables

**Figure 1 ijms-24-06907-f001:**
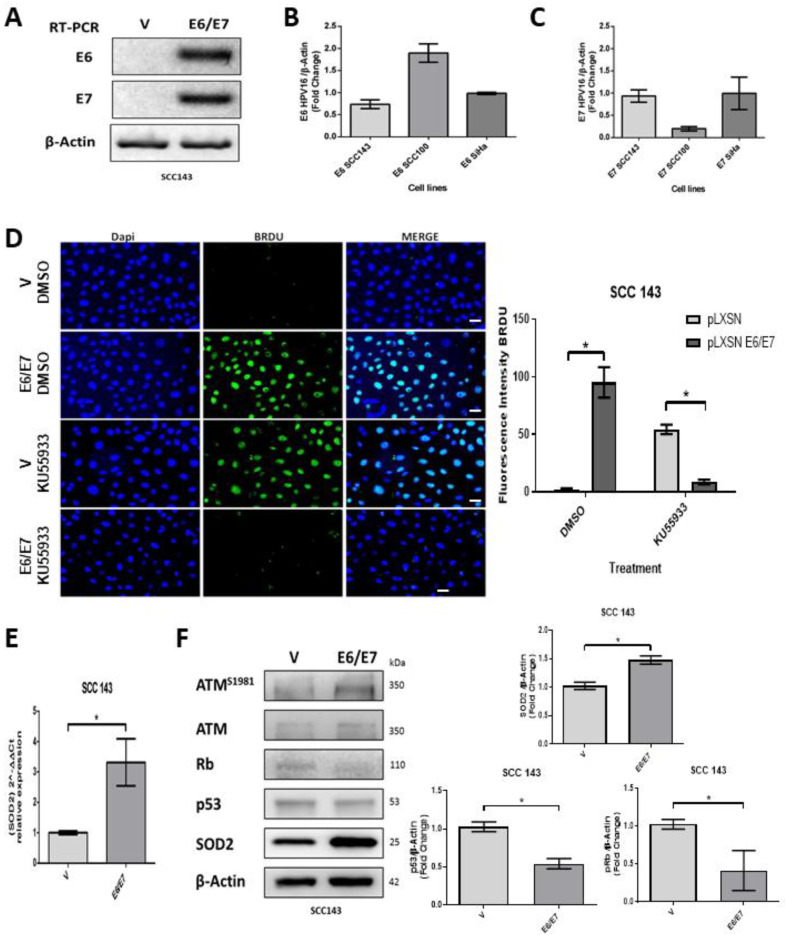
HPV16 E6/E7 induces SOD2 expression in oral cells. (**A**) HPV16 E6/E7 transcripts were evaluated by RT-PCR in SCC143V and SCC143E6/E7 cells; β-actin was used as the control. (**B**) E6 transcripts were assessed by RT-PCR in SiHa, SCC100E6/E7, and SCC143 E6/E7 cells and then normalized to the intensity of β-actin transcript. (**C**) E7 transcripts were assessed by RT-PCR in SiHa, SCC100E6/E7, and SCC143 E6/E7 cells, then normalized to the intensity of β-actin transcript. (**D**) BrdU assay to analyze DNA synthesis in SCC143V and SCC143 E6/E7 cells previously exposed to KU55933 (ATM) inhibitor. Scale bar: 10 μm. The graph represents fluorescence analysis performed with the fluorescence intensity. (**E**) The levels of SOD2 transcripts normalized by β-actin were evaluated by RT-qPCR in SCC143V and SCC143E6/E7 cells. (**F**) Western blot to evaluate ATM^S1981,^ ATM, Rb, p53, SOD2, and β-actin protein levels in SCC143V and SCC143E6/E7 cells. The graphs represent three independent Western blots for SOD2 pRb and p53 protein normalized by β-actin. Densitometric analyses were performed with ImageJ. Data are presented as the mean ± standard deviation (SD); average of three independent experiments, conducted in triplicate; * *p* < 0.05.

**Figure 2 ijms-24-06907-f002:**
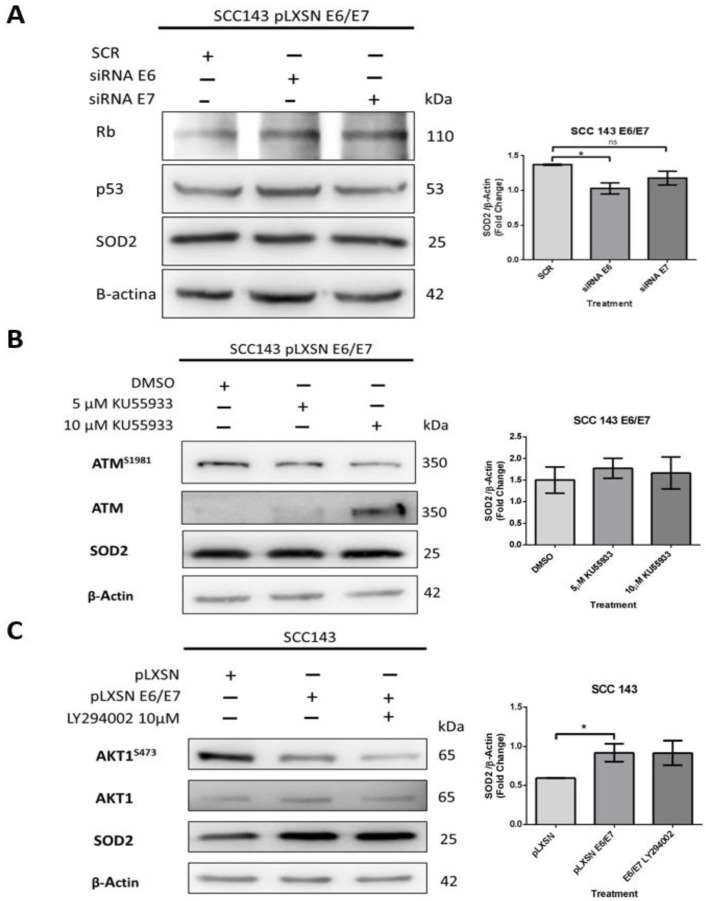
SOD2 is regulated by HPV16 E6 in an AKT1- and ATM-independent manner in oral cells. (**A**) Western blot performed for total protein extract of SCC143E6/E7 cells previously transfected with control siRNA (SCR), siRNA E6, or siRNA E7 (48 h) to evaluate Rb, p53, SOD2, and β-actin protein levels. (**B**) Western blot was performed with protein extracts from SCC143E6/E7 cells previously exposed to KU55933 (ATM) inhibitor for 24 h. The levels of total ATM, ATM^S1981^, SOD2, and β-actin used as load control were analyzed. (**C**) Western blot was performed with protein extracts from SCC143E6/E7 cells previously exposed to LY294002 (PI3K) inhibitor for 24 h. The levels of total AKT1, AKT1^S473^, SOD2, and β-actin were evaluated. The graphs represent three independent Western blots for SOD2 normalized by β-actin. Densitometric analyzes were performed with ImageJ. Data are presented as the mean ± standard deviation (SD); average of three independent experiments, conducted in triplicate; * *p* < 0.05; ns, not significant.

**Figure 3 ijms-24-06907-f003:**
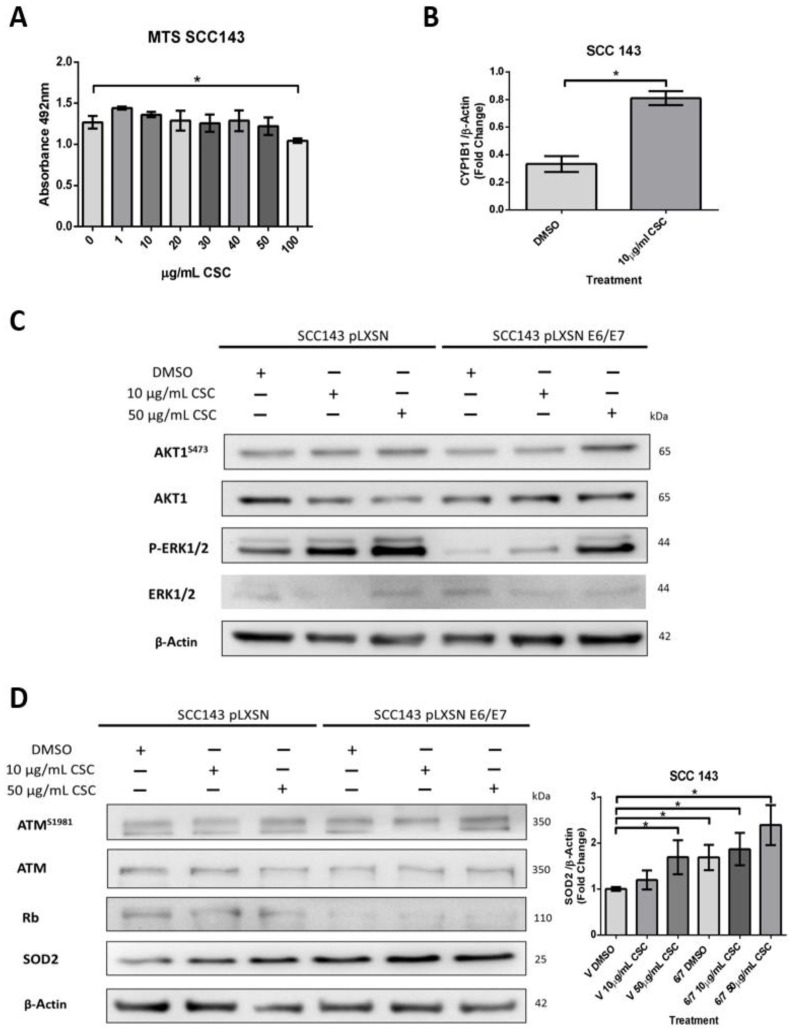
HPV16 E6/E7 oncoproteins and CSC increase SOD2 levels in oral cells. (**A**) MTS assay in SCC143 cells treated with CSC at different concentrations and incubated for 72 h. (**B**) CYP1B1 transcripts were evaluated by RT-PCR in SCC143 cells previously exposed to the inhibitor KU55933 (ATM) for 24 h. (**C**) Western blot performed for total protein extract of SCC143V and SCC143E6/E7 cells previously exposed to CSC for 24 h. The levels of total AKT1, AKT1^S473^, ERK1/2, p-ERK1/2, and β-actin were evaluated. (**D**) Western blot performed for total protein extract of SCC143V and SCC143E6/E7 cells previously exposed to CSC for 24 h. The levels of total ATM, ATM^S1981^, Rb, SOD2, and β-actin were evaluated. The graphs represent three independent Western blots for SOD2 normalized by β-actin. Densitometric analyzes were performed with ImageJ. Data are presented as the mean ± standard deviation (SD); average of three independent experiments, conducted in triplicate; * *p* < 0.05.

**Figure 4 ijms-24-06907-f004:**
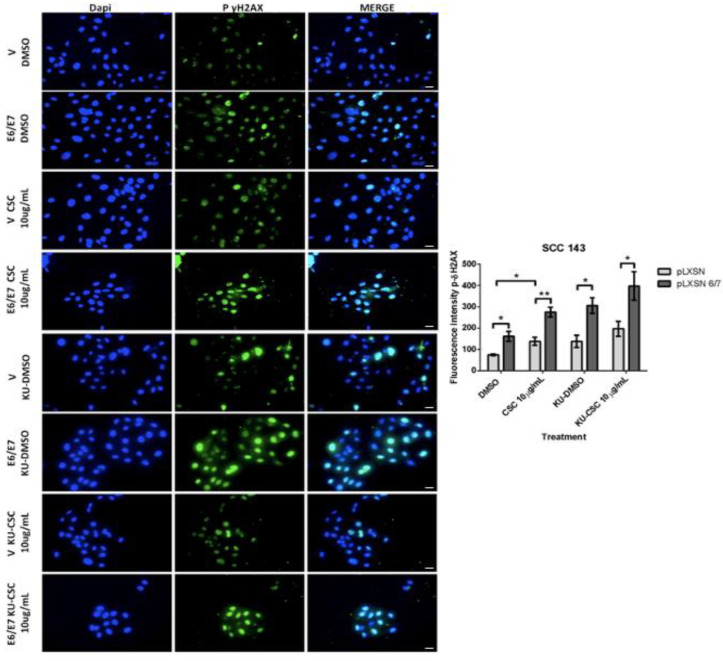
HPV16 E6/E7 oncoproteins and CSC increase DNA damage in oral cells. Indirect immunofluorescence (IFI) performed in SCC143V and SCC143E6/E7 cells previously exposed to CSC or CSC+ KU55933 (ATM) inhibitor for 24 h to evaluate γ-H2AX protein. Scale bar: 10 μm. The graph represents fluorescence analysis performed with the analysis of fluorescence intensity. Data are presented as the mean ± standard deviation (SD); average of three independent experiments, conducted in triplicate; * *p* < 0.05 and ** *p* < 0.01.

**Figure 5 ijms-24-06907-f005:**
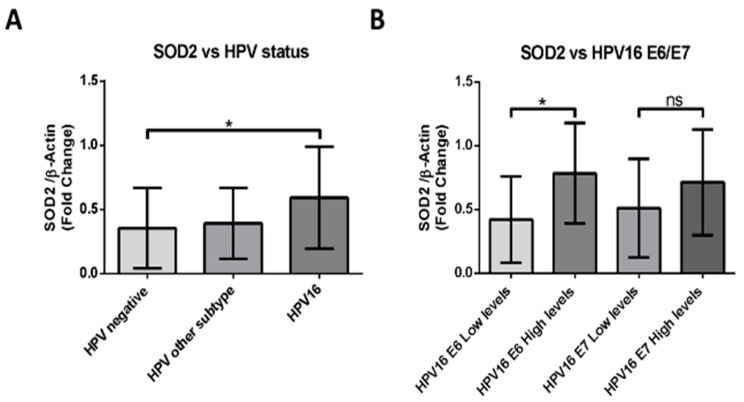
HPV16-positive HNSCCs correlate with upregulation of SOD2 transcripts. (**A**) SOD2 transcripts were evaluated by RT-PCR and normalized to β-actin transcript intensity. The sample data were separated into three groups: HPV negative, HPV16 positive, and other HPV genotypes. (**B**) The levels of the SOD2 transcript normalized by b-actin were evaluated by RT-PCR, according to the levels of HPV16 E6 or E7 transcripts. To generate these groups, the levels of E6 and E7 transcripts were evaluated in HPV16-positive samples, and they were stratified into high or low expression levels, considering the median of the data. Densitometric analyzes were performed by ImageJ software. Data are presented as the mean ± standard deviation (SD); average of three independent experiments, conducted in triplicate; * *p* < 0.05; ns, not significant.

**Figure 6 ijms-24-06907-f006:**
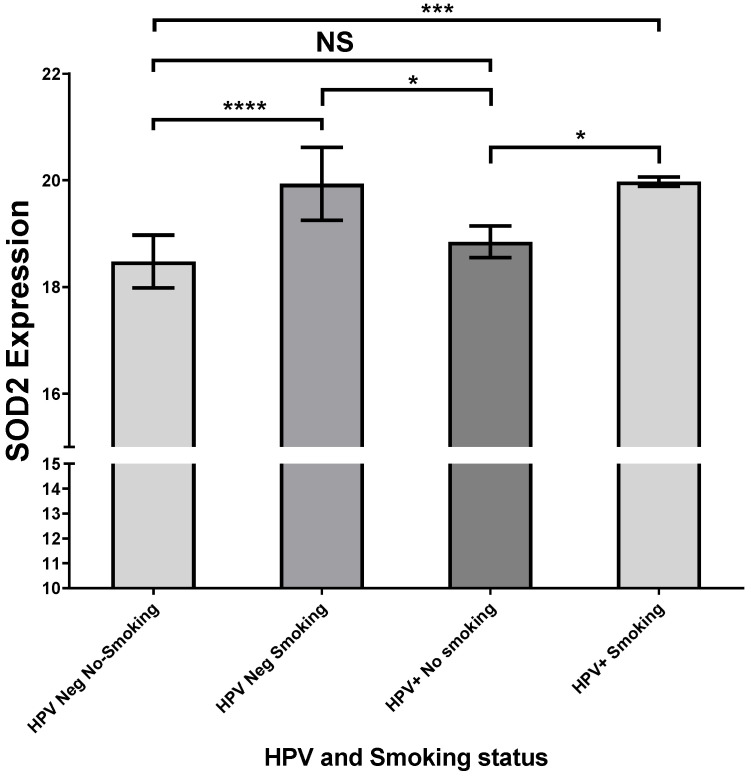
HPV positive and smoking since floor of mouth cancer present an up-expression of SOD2. Six hundred thirteen HNSSC samples were selected from the GDC TCGA Head and Neck Cancer Database, using the UCSC Xena web source. Twenty-eight HNSSC were selected under the following criteria: (1) HPV status information, (2) tobacco smoking history, (3) SOD2 expression, and (4) tumor primary site (floor of mouth); * *p* < 0.05, *** *p* < 0.001, **** *p* < 0.0001, NS, not significant.

**Figure 7 ijms-24-06907-f007:**
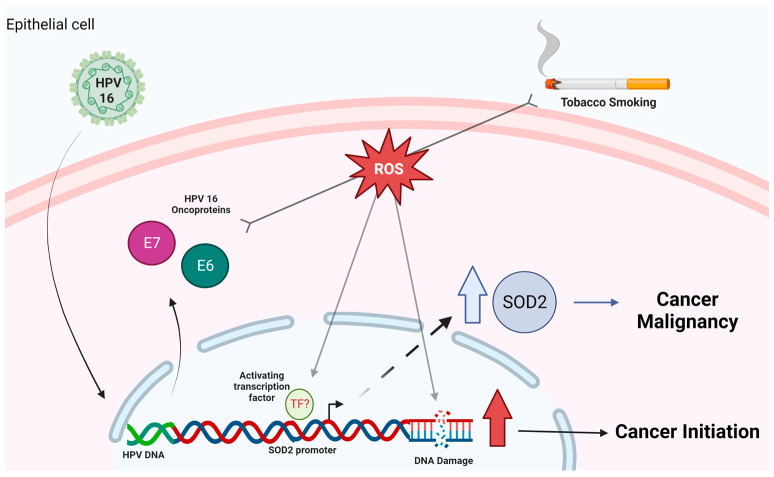
Proposed model of cigarette smoke and HPV16 interaction in HNSCC cells. HPV 16 E6 and E7 oncoproteins collaborate with cigarette smoke to increase genetic damage and SOD2 overexpression, promoting cancer initiation and malignancy, respectively. Created by BioRender.com. We allow the use of this image for educational use.

**Table 1 ijms-24-06907-t001:** Primer list.

Primer	Forward 5′-3′	Reverse 5′-3′	Size (bp)
E6 small 16	CTGCAAGCAACAGTTACTGCGA	TCACACACTGCATATGGATTCCC	96
E7 small 16	CAATATTGTAATGGGCTCTGTCC	ATTTGCAACCAGAGACAACTGAT	120
PCO3/PCO4	ACACAACTGTGTTCACTAG	CAACTTCATCCACGTTCACC	110
GP5+/GP6+	TTTGTTACTGTGGTAGATATCAC	GAAAAATAAACTTAAATCATATTC	155
SOD2	GCCCTGGAACCTCACATCAAC	CAACGCCTCCTGGTACTTCTC	111
b-actin	CCACACAGGGGAGGTGATAG	CCACACAGGGGAGGTGATAG	115

## Data Availability

Supporting data can be obtained through direct communication with the corresponding author Francisco Aguayo.

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
