# Peer review of "Interaction between Cigarette Smoke and Human Papillomavirus 16 E6/E7 Oncoproteins to Induce SOD2 Expression and DNA Damage in Head and Neck Cancer"

_ijms, 2023, doi:10.3390/ijms24086907_

Round 1
Reviewer 1 Report
In this article, the authors discussed the co-occurrence of tobacco smoking and high-risk human papillomavirus (HR-HPV) infection as risk factors for developing head and neck cancer (HNC). The study suggests that superoxide dismutase 2 (SOD2) can be independently regulated by tobacco and HPV, increasing adaptation to oxidative stress and tumor progression. The researchers analyzed SOD2 levels and DNA damage in oral cells expressing HPV16 E6/E7 oncoproteins and exposed to cigarette smoke condensate (CSC). They found that the combination of HPV and tobacco smoke exposure synergistically increased SOD2 levels and DNA damage, contributing to the development of a different clinical entity of HNC. However, there are some remaining questions to be answered:
1, Fig 1D, KU55933 (ATM inhibitor) increased cell proliferation in SCC cells. Does ATM inhibition activate cell proliferation?
2, Fig 1F, could the author perform quantitative analysis on p53 and pRb? Have the authors examined the expression of Chk2 or pChk2 which is the downstream gene of ATM?
3, Fig 2B, 10uM KU55933 increased ATM expression?
4, Fig 3C, CSC inhibited the AKT1 expression in SCC143 pLXSN as dose dependent manner, while increasing the AKT1 expression in SCC143 pLXSN E6/E7. Could the authors explain the discrepancy between two cell lines?
5, have the authors performed statistical analysis among groups, not just compared with control group? For instance, Fig 3D, and Fig 6.
6, What is the criteria for distributing the data in groups of high or low levels of E6/E7 transcript?
7, Have the authors thought about to overexpress E6 or E7 separately in SCC cell line and evaluate their conclusion that E6 upregulated SOD2?
Author Response
Dear reviewer, we appreciate all the reviews, and here are our responses.
1, Fig 1D, KU55933 (ATM inhibitor) increased cell proliferation in SCC cells. Does ATM inhibition activate cell proliferation?
R: This is an excellent observation. Decreased ATM in tumor cells is associated with increased genetic damage and apoptosis (PMID: 31299316; PMID: 24003211), which could lead to increased BrdU labelling. Considering the above, the labeling of SCC143/V cells would not be associated with an increase in proliferative capacity but rather with an increase in DNA breaks that would generate free 3OH ends.
2, Fig 1F, could the author perform quantitative analysis on p53 and pRb? Have the authors examined the expression of Chk2 or pChk2 which is the downstream gene of ATM?
R: We appreciate the comment. Analyzes were performed and included in the manuscript. We tried to test the levels of chk2 and pChek2. However, we could not obtain an antibody label which was kindly provided.
3, Fig 2B, 10uM KU55933 increased ATM expression?
R: Basal ATM levels can be upregulated in different contexts, such as increased genetic damage (PMID: 18224251). However, this behavior appears to be a classic compensatory mechanism for the cell to recover active or phosphorylated ATM levels.
4, Fig 3C, CSC inhibited the AKT1 expression in SCC143 pLXSN as dose dependent manner, while increasing the AKT1 expression in SCC143 pLXSN E6/E7. Could the authors explain the discrepancy between two cell lines?
R: This is an excellent analysis and we appreciate this comment. In SCC143 pLXSN cells, CSC activity produces a regular decrease or depletion of total AKT1 levels as its ubiquitination is increased and the phosphorylated form is increased by activating signaling pathways upstream (PMID: 21778238; PMID: 12511591). On the other hand, the levels of AKT1 in its total form are downregulated in SCC143 pLXSN E6/E7 cells, which could be compensated by the presence of CSC in different concentrations since this product exerts multiple molecular changes in the cell. However, we do not investigate these changes further because we took only these proteins as a functional control of CSC.
5, have the authors performed statistical analysis among groups, not just compared with control group? For instance, Fig 3D, and Fig 6.
R: We add the statistical differences found.
6, What is the criteria for distributing the data in groups of high or low levels of E6/E7 transcript?
R: An analysis of the data obtained by RT-PCR with the Image J program was performed, these data normalized with the respective b-actin were organized to calculate the median of the data obtained, and all data greater than or equal to the median were considered high expression and those below the median were considered low expression.
7, Have the authors thought about to overexpress E6 or E7 separately in SCC cell line and evaluate their conclusion that E6 upregulated SOD2?
R: Soon we want to carry out a further study to evaluate SCC143 cells that express the oncoproteins independently. In this study, we want to elucidate molecularly how both factors (E6 HPV16-CSC) regulate SOD2 levels. Unfortunately, we do not have cells that stably express E6 and E7 separately.

Reviewer 2 Report
This study showed that oral cells expressing HPV16 E6/E7 oncoproteins exposed to cigarette smoke concentrate (CSC) synergistically increased SOD2 levels and DNA damage which predispose to carcinogenesis.
Previous studies showed a synergistic effect between smoking and both HPV-16 status and HPV-16 viral load in cervical cancer development. Also studies showed that CSCs prepared from cigarettes that primarily heat tobacco (Eclipse) were much less cytotoxic than those prepared from Kentucky 1R4F reference cigarettes. This is of particular interest in the context of this study, considering applied CSC concentrations and the effect on the cells independent of the synergistic effect with HR HPV.
Therefore it would be worth adding in this study the source and preparation of CSC, as well as the CSC nontoxic concentrations used as this is not stated. The question also remains whether activity of tested parameters in cells, which, as shown, depends on different concentrations of CSC, is time-dependent after exposure and in which time range?
Since authors use alternately the term tobacco smoke and cigarette smoke, suggestion is to use more precise term cigarette smoke all over the text considering that cigarette smoke concentrate was used in the study.
Suggestion is also to reformulate the title and to use the term Interaction or Synergistic effect ..instead of the word „Cooperation“, if authors agree.
Please provide the reference or state the permission to reuse Figure 7.
I recommend publishing the paper with amendments as suggested.
Author Response
Dear reviewer, we appreciate all the reviews, and here are our responses.
1-Therefore it would be worth adding in this study the source and preparation of CSC, as well as the CSC nontoxic concentrations used as this is not stated. The question also remains whether activity of tested parameters in cells, which, as shown, depends on different concentrations of CSC, is time-dependent after exposure and in which time range?
R: What is indicated is an excellent analysis that we had not yet perceived. Therefore, we add a sentence explaining what is indicated in materials and methods.
2-Since authors use alternately the term tobacco smoke and cigarette smoke, suggestion is to use more precise term cigarette smoke all over the text considering that cigarette smoke concentrate was used in the study.
R: We appreciate this recommendation. we corrected the term tobacco smoke where appropriate.
3-Suggestion is also to reformulate the title and to use the term Interaction or Synergistic effect ..instead of the word „Cooperation“, if authors agree.
R: We agree with what has been stated. We change the word cooperation to interaction in the title.
4-Please provide the reference or state the permission to reuse Figure 7.
R: We add the permission.

Reviewer 3 Report
The authors of this study investigated SOD2 levels and DNA damage in oral cells while also demonstrating that the interaction of cigarette smoke with the HPV16 E6 and E7 oncoproteins causes SOD2 levels and genetic damage in oral cells. The paper is well-written and a great starting point for further study. As a side note, the discussion needs to mention the study's constraints. I'd like to congratulate the authors and wish them luck with their upcoming work!
Author Response
Dear reviewer, we appreciate all the reviews, and here goes our answer.
1-The authors of this study investigated SOD2 levels and DNA damage in oral cells while also demonstrating that the interaction of cigarette smoke with the HPV16 E6 and E7 oncoproteins causes SOD2 levels and genetic damage in oral cells. The paper is well-written and a great starting point for further study. As a side note, the discussion needs to mention the study's constraints. I'd like to congratulate the authors and wish them luck with their upcoming work!
R: We appreciate this comment, and it is a great pleasure to hear this kind of motivation from a reviewer. We will continue working to continue with this type of study in the future. We added the limitations of the study.
